# What Happens after Hospital Discharge? Deficiencies in Medication Management Encountered by Geriatric Patients with Polypharmacy

**DOI:** 10.3390/ijerph18137031

**Published:** 2021-06-30

**Authors:** Laura Mortelmans, Elyne De Baetselier, Eva Goossens, Tinne Dilles

**Affiliations:** 1Department of Nursing Science and Midwifery, Centre for Research and Innovation in Care (CRIC), Nurse and Pharmaceutical Care (NuPhaC), Faculty of Medicine and Health Sciences, University of Antwerp, 2610 Antwerp, Belgium; elyne.debaetselier@uantwerpen.be (E.D.B.); eva.goossens@uantwerpen.be (E.G.); tinne.dilles@uantwerpen.be (T.D.); 2Department of Public Health and Primary Care, KU Leuven, 3000 Leuven, Belgium; 3Research Foundation Flanders (FWO), 1000 Brussels, Belgium; 4Department of Patient Care, Antwerp University Hospital (UZA), 2610 Antwerp, Belgium

**Keywords:** medication management, self-management, medication use, polypharmacy, older people

## Abstract

This study aimed to describe post-discharge medication self-management by geriatric patients with polypharmacy, to describe the problems encountered and to determine the related factors. In a multicenter study from November 2019 to March 2020, data were collected at hospital discharge and two to five days post-discharge. Geriatric patients with polypharmacy were questioned about medication management using a combination of validated (MedMaIDE) and self-developed questionnaires. Of 400 participants, 70% did self-manage medication post-discharge. Patients had a mean of four different deficiencies in post-discharge medication management (SD 2.17, range 0–10). Knowledge-related deficiencies were most common. The number of medicines and the in-hospital provision of medication management by nurses were significant predictors of post-discharge medication management deficiencies. In addition to deficiencies in knowledge, medication-taking ability and obtaining medication, non-adherence and disrupted continuity of medication self-management were common in geriatric patients with polypharmacy post-discharge. Improvements in in-hospital preparation could avoid medication self-management problems at home.

## 1. Introduction

Medication management capacity (MMC) can be defined as “the cognitive and functional ability to self-administer a medication regimen as it has been prescribed” [1]. Medication self-management requires patients to know their medication, to understand their medication schedule, to be able to purchase medication and to have the functional skills such as correctly identifying medications, opening containers, and selecting the proper dose and time of administration [2].

The term medication self-management refers to the patient’s need to fulfill a set of actions in order to manage medicines [3]. The process starts with fulfilling a prescription. During the second step, patients should learn how to use medicines in a safe and correct manner which requires medication knowledge. Organizing medication intake and planning daily medication schedules is the third step. The fourth step comprises of actually taking the prescribed medication. Monitoring medication intake and evaluating possible effects or symptoms related to medicines in order to undertake any action if needed is described as the fifth step. The last step concerns the act of sustaining a correct medication intake routine in a safe and appropriate way [3,4]. However, the literature suggests individuals struggle in various ways during the completion of this set of actions influencing medication adherence [5,6,7,8,9,10]. Adherence can be defined as “the process by which patients take their medications as prescribed” [11].

During hospitalization, medication management is usually performed by healthcare professionals, although after discharge, patients have to manage their medicines themselves, often with limited guidance from healthcare professionals [12]. If support is lacking, shortcomings in medication self-management can lead to non-intentional therapy non-adherence. About 24% to 40% of patients are found to be non-adherent to their medication regimen after hospital discharge [5,13]. It is crucial to detect and address self-management problems prior to hospital discharge in order to promote adherence and safe medication use at home, especially in the elderly [14].

Geriatric patients are at risk of medication self-management problems because of functional and cognitive decline and geriatric syndromes that accompany aging [15]. In addition, aging increases the risk of polypharmacy. Polypharmacy is commonly defined as “the minimum concomitant intake of at least five medicines” [16,17]. Prevalence rates of polypharmacy in Flanders, Belgium for patients aged 75 years and older increased significantly by 1.3% per year between 2011 and 2015 [18]. Furthermore, polypharmacy is associated with drug-related problems such as side effects, interactions and medication errors [19]. Moreover, pharmacokinetic and pharmacodynamic changes make the elderly more vulnerable to adverse events from aberrant medication intake [20]. Unfortunately, previous studies on medication self-management are scarce and have not focused on geriatric patients after hospital discharge [21,22,23]. Therefore, the results of these studies are not transferable to this subpopulation.

This study aimed (1) to evaluate post-discharge medication self-management and the problems encountered by geriatric patients with polypharmacy and (2) to identify the factors associated with post-discharge medication management deficiencies.

## 2. Materials and Methods

### 2.1. Design

A multicentre, quantitative cross-sectional observational design was used to investigate medication management in geriatric patients with polypharmacy. A two-part survey was administered between November 2019 and March 2020. The first part of the questionnaire was administered at the day of hospital discharge. Two to five days after discharge, a home visit was scheduled for the completion of the second part of the questionnaire that focused on medication management.

### 2.2. Participants and Setting

Twelve general hospitals in Flanders, Belgium participated. At each hospital, patient selection was initiated from a convenience sample of wards, based on the availability of eligible patients. Hospitalized patients were eligible if they were at least 75 years old, used five or more prescribed medicines at the time of hospital discharge (i.e., polypharmacy) and spoke Dutch. Patients receiving palliative care (i.e., palliative status, palliative file, palliative care pathway), having a neuropsychiatric diagnosis of dementia, being legally incapable and patients who did not return home after discharge were excluded. Patients’ eligibility, based on the aforementioned criteria, was evaluated by the (head)nurse of the ward. All eligible patients from the selected wards who were willing to participate were included during the data collection period until the desired sample size was reached (consecutive sampling).

The sample size was calculated prior to the study, using the Sampsize calculator for prevalence studies [24]. Precision (%), prevalence (%), target population size and confidence level were entered. To reach the highest estimated sample size, an infinite population size (defined by the number 0) and an estimated prevalence of 50% (default value) was used. To ensure the validity of the data with a precision of 5%, an estimated prevalence of self-management problems of 50%, an unknown population size and a 95% confidence interval, the desired sample size was 385. Due to the maxim of n = 40 × number of predictors, stepwise multiple logistic regression was allowed with this sample size to determine the factors related to post-discharge medication management deficiencies [25].

### 2.3. Data Collection

Data collection proceeded in several steps. Firstly, eligible patients were invited to participate in the study during hospitalization. If patients refused to participate, the reasons for a non-response were surveyed. Secondly, patients willing to participate were visited in hospital at the day of discharge. A questionnaire on socio-demographic data was administered, a geriatric risk profile was calculated, the patients’ medication schedule was copied to collect data about patients’ medication use and a home visit was scheduled two to five days after discharge. Thirdly, during this home visit, a questionnaire about medication self-management and medication management deficiencies was administered. Trained research assistants, who were all nurses with a Bachelor’s degree, collected data between November 2019 and March 2020. Through an information session, the research assistants were informed about the administration of the questionnaire upon discharge and during the home visit.

At the day of discharge, patients were informed about the fact that the study was focusing on discharge management in the broad sense in order to avoid research bias. During the home visit, the fact that the study focused on medication management in the context of discharge management was clarified.

#### Measurements

To describe the population, the following data were collected: age, gender, educational level, type of hospitalization, reason for hospitalization, length of hospital stay, chronic diseases and support to reside at home. Patients were screened for geriatric care problems through a calculation of a geriatric risk profile, based on the Triage Risk Screening Tool [26]. The latter was scored as follows: cognitive impairment present (2 points), living alone or no help by partner/family possible (1 point), difficulties with transfers in the past six months (1 point), hospitalized within the past three months (1 point) and taking five or more medicines (1 point). A score equal to or higher than 2/6 meant that the patient had a positive geriatric risk profile. The lowest possible score was 1 as each participant received 1 point for polypharmacy.

The data on medication management capacity were collected using the Medication Management Instrument for Deficiencies in the Elderly (MedMaIDE) [27]. This validated instrument consists of three areas important for proper medication management: (1) medication knowledge (8 questions), (2) functional ability to take medication (5 questions) and (3) obtaining medication (6 questions). MedMaIDE items were dichotomous responses that could be scored Yes (0) and No (1), respectively. Although the MedMaIDE is comprised of 19 items, only 13 items were used to calculate the deficiency sub scores (per area) and the overall deficiency score. These 13 items are considered critical by Orwig et al. [27], as they are crucial for the self-management of medication. These consisted of five medication knowledge-related items, five items related to medication-taking ability and three items related to obtaining medication. Therefore, sub scores related to medication knowledge and medication-taking ability ranged between 0 and 5 points while the deficiency sub score related to obtaining medication varied between 0 and 3. The total medication management deficiency score had a maximum of 13. The MeDMaIDE uses dichotomous variables to collect data on medication knowledge, which only provides information on whether or not knowledge is present. To gather information on the extent to which a patient has knowledge of his medication, the number of medicines for which the patient knew the name, indication, dose, route of administration and time of intake were noted. Medication knowledge was objectively assessed by research assistants for all the patients’ medicines based on the medication schedule. Patients were not allowed to use their own medication schedule or notes to answer questions related to medication knowledge.

In-hospital medication management (i.e., storing/preparing/administering medication), discharge policy (i.e., in-hospital preparation for medication self-management and medication-related information at discharge), post-discharge management of side effects by patients, medication changes after discharge and correct medication intake after discharge were surveyed using four- and five-point Likert scales, 10-point rating scales and self-developed multiple-choice questions.

### 2.4. Data Analysis

Data were analysed using SPSS 27.0. Categorical variables were described using frequency distributions while mean and standard deviations were used for continuous variables. The normality of the data was tested using absolute skewness and kurtosis [28]. The Little’s MCAR test was used to assess missing values which were found to be missing completely at random (MCAR) [29]. Listwise deletion was used to treat missing cases in each sub analysis. Paired analyses were performed to determine the differences in medication management before and after hospitalization using McNemar statistics. The differences between patients who did fully self-manage (preparing and administering medicines independently) and those who did not fully self-manage their medication after discharge, were calculated using parametric statistics (chi-squared tests for nominal and independent t-tests for continuous variables). To determine which factors influenced medication management deficiencies after hospital discharge, a stepwise logistic regression analysis was conducted. Four binary outcome variables were created: (1) medication management deficiencies in general, (2) medication-related knowledge deficiencies, (3) deficiencies related to medication-taking ability and (4) deficiencies related to obtaining medication. Based on the total deficiency score and sub scores of the MedMaIDE, patients were subdivided into patients with (deficiency score of at least 1) and without deficiencies (deficiency score of 0). First, a univariate analysis was performed to determine which factors were significant predictors of medication management deficiencies. Subsequently, only significant variables from the univariate analysis were included in the multiple logistic regression. Multicollinearity was assessed using correlation matrices. A *p*-value < 0.05 was considered statistically significant.

## 3. Results

### 3.1. The Research Population

A total of 469 patients were invited to participate in the survey of which 69 refused (14.7%). Reasons for non-participation were not willing to spend time on research (n = 17), did not want to share personal data (n = 7), did not want a home visit (n = 25), felt too weak/too sick (n = 8), and other reason(s) (n = 7). Ten patients did not want to provide an explanation for non-response.

Four hundred participants completed the survey (85.3%). The characteristics of the research population are shown in Table 1. The mean age of the participants was 82 years [SD 5.0] and 53% were women. Only 25% had an educational level higher than level four of the European Qualification Framework [30]. About half of the sample (49%) spent more than seven days in hospital. Approximately 8% were admitted for medication review. Most of the participants (91%) had a positive geriatric risk profile. The participants had a mean of two chronic diseases and took a mean of nine different prescribed medications at discharge. Moreover, 62% needed support to reside in their own home.

About 81% of patients reported at least one medication change during the six months prior to their last hospital admission. Furthermore, medication changes after hospital discharge were frequent: new medication was started in 58%, medication was stopped in 18%, dose changes occurred in 13% and changes in time of intake occurred in 5% of patients. According to the participants, the main causes for these medication changes included altered health status (74%) and different prescribers (6%).

### 3.2. Medication Management before, during and after Hospitalization

About 78% of the patients did fully self-manage their medicines before hospital admission. In hospital, this was only 13%. In 74% of the patients, nursing staff managed all medicines during hospitalization. The remaining 13% received help in preparing their medication but administered them independently. After discharge, 70% did fully self-manage their medication, 27% received help with preparing their medication but self-administered their medicines and 3% received help with preparing and administering medicines at home (See Figure 1).

Comparing medication self-management before and after hospitalization, some patients lost their ability to prepare their medication (*p* < 0.001). Taking medication independently remained unchanged (both 97%). Significantly more participants used aids such as a calendar or a pillbox after hospital discharge (58% vs. 63%, *p* < 0.001).

About 54% of patients indicated they were not prepared or insufficiently prepared during their hospital stay to self-manage their medication at home. Patients who did fully self-manage their medication after discharge were more likely to indicate they were adequately prepared during hospitalization in contrast to those who did not (52.6% vs. 31.0%; X^2^ = 4.091; *p* < 0.001). About 74% had no conversation about how to manage their medication at home. Furthermore, only 26% indicated that they received enough information about medication at time of discharge. Patients were most frequently informed about time of intake (78%) and dose (70%), while only 10% were informed about potential side effects. About 15% received no schedule at all for medication intake after discharge. When a schedule was given to the patient, in addition to the name of the drug, dose and time were indicated in 94% and 98% of patients, respectively. Medication-related information was mostly provided by nursing staff (70%). The participants who did fully self-manage after discharge received information more often from the nursing staff at discharge as compared to those who did not (74% vs. 58%; X^2^ = 6.467; *p* = 0.011) (See Table 2).

On average, participants rated their satisfaction with the help received during hospitalization to take their medication correctly after discharge 7.3/10 [SD 2.3]. Nevertheless, 62% had to ask for additional help regarding medication after discharge. Furthermore, 13% indicated it was difficult to take back responsibility for their medication management after discharge. Significantly more patients who found it difficult to take back responsibility did not fully self-manage (X^2^ = 42.321; *p* < 0.001). Medication management assistance should be extended according to 8% of self-managing patients and according to 26% who did not fully self-manage their medicines at home (X^2^ = 24.530; *p* < 0.001). Patients who did fully self-manage their medicines post-discharge were significantly more satisfied with their own medication management after discharge (t = −7.413; *p* < 0.001) (See Table 2).

#### 3.2.1. Correct Medication Intake after Discharge

Two to five days after discharge, almost 23% indicated that they had stopped their prescribed medication therapy earlier than agreed with the physician and 25% changed the way medicines should be taken based on their own knowledge and experience (See Table 2). Furthermore, approximately 20% of patients indicated that they sometimes took their medication incorrectly: 9% of them took an incorrect dose and 63% took their medication at the wrong time. The reasons given for not correctly taking medication were forgetfulness (64%), adjusting the medication regimen seems better to the patient (19%), not knowing how to do so (6%), not being able to do so (6%) and not willing to do so (3%).

#### 3.2.2. Management of Side Effects

Approximately 17% of patients reported that they experienced side effects of their medication after discharge. The most common side effects were nausea/vomiting (28%), followed by dizziness (15%), drowsiness/sedation (14%), constipation (12%) and skin rash/itching (12%). Of the participants who experienced side effects, 23% did nothing while 8% (temporarily) stopped taking their medication. Furthermore, 14% reported side effects to the home care nurse and 55% reported these to the general practitioner.

#### 3.2.3. Medication Management Deficiencies after Discharge

Approximately 90% of patients experienced at least one medication management deficiency after discharge. The deficiency scores are shown in Figure 2. Patients had a mean of four different deficiencies in medication management (range 0–10). Most deficiencies were related to medication knowledge (mean 3.1 [SD 1.8]). About 86% experienced at least one knowledge-related deficiency. On average, patients knew the name of the medication for 55% of their prescribed medicines. Furthermore, they could correctly identify the time of intake for an average of 70% of their medicines, the route of administration for 84%, the indication for 62% and the dose for 44% of their medicines. Only 26% of patients knew the name of all the medicines they had to take. Furthermore, 57% of patients could not correctly identify the time of intake for all medicines and 40% could not correctly indicate the route of administration for all medicines. In addition, 66% did not know the indication and 77% did not know the correct dose of all prescribed medicines.

Deficiencies related to the functional ability required to take medication (mean 0.2 [SD 0.6]), as well as deficiencies in obtaining medication (mean 0.4 [SD 0.6]) were less common. However, 2% of patients reported to be unable to fill a glass of water to take their medicines, 11% were unable to open the medication container (e.g., vial, pill box), 4% were unable to bring their medication up to their mouth, 3% were unable to count the number of pills needed and 3% were unable to swallow medication.

In terms of obtaining medication, almost half of patients (48%) did not have all the required medicines available at home after discharge. Furthermore, 31% indicated that they had not required prescriptions and 3% did not know who to contact for a new prescription. In addition, 3% indicated they did not have the resources to obtain medication (e.g., cannot arrange transportation to pharmacy, pharmacy does not deliver medication at home, no help from others to pick up medication).

Patients who fully self-managed their medicines after discharge experienced less knowledge-related deficiencies (mean 2.8 [SD 1.9] vs. mean 3.9 [SD 1.5]; t = 5.924; *p* < 0.001), less medication-taking ability deficiencies (mean 0.1 [SD 0.3] vs. mean 0.6 [SD 0.9], t = 6.003; *p* < 0.001) and, therefore, less medication management deficiencies in general (mean 3.2 [SD 2.1] vs. mean 4.9 [SD 1.9]; t = 7.642; *p* < 0.001) compared to those who received help with their medication management after discharge. No differences were found in deficiencies related to obtaining medication (t = 1.027; *p* = 0.306).

### 3.3. Factors Influencing Medication Management Deficiencies after Discharge

Using a stepwise logistic regression analysis, the factors associated with medication management deficiencies after discharge were studied. Patient and medication management related factors were considered. As shown in Table 3, patients experiencing medication management deficiencies took a larger number of medicines at discharge (mean 9.4 [SD 3.4] vs. mean 7.7 [SD 2.5]; W = 9.212; *p* = 0.002). Patients with in-hospital medication management performed by nurses experienced post-discharge deficiencies more often in medication management as compared to patients who fully/partially self-managed medication during hospitalization (76.1% vs. 23.9%; W = 8.518; *p* = 0.004). Furthermore, patients who were not prepared or insufficiently prepared in hospital to manage medication at home were more likely to have post-discharge medication management deficiencies (OR = 2.00; 95% CI [1.02, 3.94]). In the multiple logistic regression, the number of prescribed medicines at discharge (OR = 1.19; 95% CI [1.05, 1.35]) and in-hospital provision of medication management by a nurse (OR = 2.42; 95% CI [1.21, 4.82]) were found to significantly predict the occurrence of post-discharge medication management deficiencies. This model explained 9% of the variance of the respective outcome (*p* < 0.001).

Furthermore, the factors associated with the different types of deficiencies were studied. No patient or medication management characteristics were found to be significant predictors of deficiencies in obtaining medication. Compared to the multiple logistic regression analysis of medication management deficiencies in general, in-hospital medication management by nurses was no significant predictor of medication knowledge deficiencies. However, patients who were not prepared or insufficiently prepared in hospital had a greater chance of experiencing knowledge-related deficiencies (OR = 2.02; 95% CI [1.12,3.64]) (See Table 4).

As shown in Table 5, a multiple logistic regression analysis resulted in a model explaining 18.7% of the variance in deficiencies in medication-taking ability after discharge. Four predictors were found to be significant: increasing age (OR = 1.08; 95% CI [1.02,1.15]), increasing number of chronic diseases (OR = 1.28; 95% CI [1.07,1.54]), the need for help to reside in their own home (OR = 3.83; 95% CI [1.54,9.52]) and making their own decisions about medication intake in hospital (OR = 0.38; 95% CI [0.19,0.75]).

## 4. Discussion

Medication self-management comprises a wide range of tasks that individuals must successfully perform to correctly follow a prescribed medication regimen. Knowledge, functional skills and behaviors are, therefore, indispensable [3]. This study demonstrated that geriatric patients with polypharmacy experience various struggles during the process of medication self-management as they were found to have on average four different deficiencies in post-discharge medication management.

The medication self-management process starts with a prescription which has to be filled and picked up [3]. This study showed a considerable proportion of patients already experience problems at this stage as they do not possess the required prescriptions, do not know who to contact for a prescription or do not have the resources to obtain medication. A study by Marques et al. confirmed that accessing the medication prescribed upon discharge is a hurdle frequently encountered by patients [31]. Being unable to obtain medication jeopardizes the entire medication self-management process. Therefore, prior to hospital discharge, healthcare providers should assess if patients are able to obtain their medicines and if they can rely on (in)formal caregivers to purchase medication [31,32].

In addition to being able to obtain medication, patients should have knowledge and understanding of their medicines that are mandatory to ensure safe and effective use. Yet, this appears to be problematic for geriatric patients with polypharmacy as 86% experienced at least one knowledge-related deficiency. Approximately three-quarters of patients did not know the name or correct dose of their entire set of prescribed medicines and two-thirds did not know the indication of all medicines after discharge. Furthermore, 40% could not correctly identify the administration route of all medicines. This is a high percentage and a somewhat odd result, as most medicines had to be administered orally. Perhaps the phrasing of the question was too complex and patients did not understand what was meant by the administration route. However, inadequate medication knowledge was also addressed in previous research by Romero-Sanchez et al. where only 28% of patients showed adequate knowledge [7]. In our sample, in-hospital preparation was unsatisfactory, which seems to be an important predictor of post-discharge knowledge deficiencies. Although a medication schedule can be helpful in organizing how and when to take medicines within daily routines, approximately one-third did not understand the medication schedule or did not receive one at all at discharge. Furthermore, more than half of the sample needed to ask for additional help after discharge to take medicines correctly. These results emphasize the need to properly educate patients about medication names, purpose, dose, time of intake, administration route and important side effects prior to discharge [33]. Prior to discharge, medication-related information should be given both verbally and in written form [34]. Nursing staff have a pivotal role in providing such information as they are in contact with patients on a regular basis [35].

However, pharmacists can provide discharge counselling and education as well [36,37]. Since 2007, pilot projects funded by the government have been launched in more than half of the acute Belgian hospitals to implement clinical pharmacy activities, such as medication counselling at discharge [37]. Furthermore, since 2017, patients with polypharmacy can choose a home pharmacist, i.e., a community pharmacist as a reference pharmacy. The added value of a home pharmacist consists of the individualized support of these patients and the provision of an up-to-date medication schedule [38,39]. In their systematic review, Ensing et al. corroborated the important role of pharmacists to secure continuity of care in multifaceted programs across healthcare settings. Still, they also emphasized that close collaboration with physicians and nurses is essential during and after hospitalization [40]. To stimulate collaboration between the general practitioner and the pharmacist to improve the safe and rational use of medication, Medical Pharmaceutical Consultations (MPC) can be organized in Belgium. Hospital physicians and hospital pharmacists can also be involved. The main aim of the MPC is to discuss problems in practice and to provide possible solutions to improve pharmaceutical care provided to patients [41].

Although lacking medication knowledge was the most common deficiency, not being able to take medication might have a larger impact on the medication self-management process as this hinders the act of administering. A small percentage of patients experienced problems such as opening medication packages or swallowing medication. Previous research among older patients showed similar problems [8,42]. Healthcare providers should assess patients’ competencies to self-manage medication at home, taking into account not only physical but also mental conditions [43,44]. If patients are unable to self-manage their medicines at home, assistance from (in)formal healthcare providers appears to be indispensable.

Considering post-discharge medication management deficiencies in general, in-hospital medication management by nursing staff was a significant predictor. During hospitalization, nurses took over patient’s medication management in 74% of cases, resulting in a disruption of the continuity of medication self-management. In daily practice, allowing more patients to self-manage medication during their hospital stay might be beneficial. Preliminary research showed an improvement in self-care competences, medication knowledge, adherence rates, patient satisfaction and patient safety as a result of in-hospital medication self-management [43,45,46].

According to Bailey et al., problems with medication management may affect medication adherence [3]. In this study, a significant proportion of patients forgot to take their medicines, took medicines at the wrong time, administered an incorrect dose or were not physically able to take medication (i.e., non-intentional non-adherence). Furthermore, 23% of patients stopped their medication therapy earlier than allowed by their prescription, for example because of side effects (i.e., intentional non-adherence). An interpretation of this high intentional non-adherence rate, two to five days after discharge, should, however, be performed with caution. Participants may have reported medication non-adherence over a longer period rather than only medication non-adherence after hospital discharge. Nevertheless, the study by Mulhem et al. indicated a higher non-adherence rate (43%) 24 to 48 h after hospital discharge [47]. Non-adherence can result in poor health outcomes (e.g., increased mortality, decreased quality of life, loss of productivity), increasing healthcare service utilization and healthcare expenditures [48,49,50,51,52]. To address medication non-adherence at home, interventions should focus on identifying and addressing the abovementioned deficiencies prior to discharge. Healthcare professionals should improve patients’ self-management capacity (i.e., physical and cognitive ability) and should involve the support of (in)formal caregivers if necessary [6].

### 4.1. Implications for Practice

Healthcare providers should identify and address medication management deficiencies prior to hospital discharge as a strategy to avoid the development of problems at home. Therefore, patients’ individual ability to self-manage a medication regimen after hospital discharge should be assessed, taking into account physical and mental conditions. Various tools are available to assess medication management capacity in older adults such as the Drug Unassisted Grading Scale, the MedTake Test or the MedMaIDE [53].

If patients appear to be unable to obtain their medicines themselves, several possibilities should be considered in consultation with the patient such as delivery of medicines by a pharmacy, an informal carer or home care nurse who picks up prescriptions/medicines or a regular home visit by the general practitioner to provide the patient with the necessary prescriptions. Prior to discharge, opportunities to address medication-taking ability deficiencies should be explored as well, such as supplying tools designed to help open specific packaging forms [54].

To address medication knowledge-related deficiencies, healthcare professionals should pay attention to whether geriatric patients understand their medication and intake instructions. Medication self-management support by healthcare providers should be improved. Interventions, such as providing medication related education prior to discharge, should be clearly described in terms of content and frequency as such information is currently lacking [45,46,55,56,57].

Furthermore, allowing more patients to self-manage medicines during a hospital stay would provide an opportunity to sustain continuity in patients’ medication self-management process, detect problems related to medication self-management and increase patients’ self-management capacity. Healthcare providers (i.e., nurses, pharmacists and physicians) can support patients during this process, providing education and counselling. In-hospital medication self-management, under supervision of healthcare providers, will make patients feel more empowered to self-manage medicines after discharge.

### 4.2. Strengths and Limitations

This is the first study examining home medication experience issues and factors related to medication management deficiencies after hospital discharge in a geriatric population with polypharmacy. This study had a high participation rate as 85% of all eligible patients completed the survey. The impact of the non-response bias is, therefore, rather small. To detect deficiencies in medication management, a validated instrument suited for the elderly was used [27].

However, this study had some limitations. Firstly, the use of self-reported data may be subjective to desirability and recall bias. Secondly, although there was an almost equal proportion of male and female participants (47% vs. 53%) which positively affects generalizability of the study results, other demographical variables (e.g., country) and the use of convenience sampling will limit generalizability. Thirdly, the survey was administered by different research assistants. Despite the fact that training was provided on the methodology used to collect data, the uniformity of the data collection may be disputable. Fourthly, patients were questioned about medication changes after discharge. Based on the high prevalence of medication changes two to five days post-discharge, we suspect participants could have misinterpreted this question. It is likely that patients compared medication before and after hospitalization and indicated, based on this comparison, whether there were any medication changes two to five days after discharge. Therefore, we should be cautious about these results. Fifthly, some questions allowed patients to fill in multiple answers, which resulted in multiple dichotomous variables. Instead of defining multiple response sets, each possible answer was treated as a separate dichotomous variable in the analysis and chi-squared tests were used. However, since the answers of the row variables were not completely independent of each other, the chi-square statistic might have inflated. Lastly, we focused on medication management in patients with polypharmacy (i.e., complex pharmacotherapy) without considering the differences in medication management between different medication(classes). This can be seen as a limitation of the study as some medications (classes) are more difficult to manage than others with a possible impact on experiencing medication management deficiencies. Therefore, we recommend that this aspect is included in subsequent research.

## 5. Conclusions

This study identified problems related to post-discharge medication self-management in geriatric patients with polypharmacy including deficiencies in obtaining medication and medication-taking ability, lack of medication knowledge, non-adherence and disrupted continuity of patients’ medication self-management. During hospitalization, healthcare providers should identify and address medication self-management issues. To prevent problems after hospital discharge, in-hospital preparation should be improved to support medication self-management at home.

## Figures and Tables

**Figure 1 ijerph-18-07031-f001:**
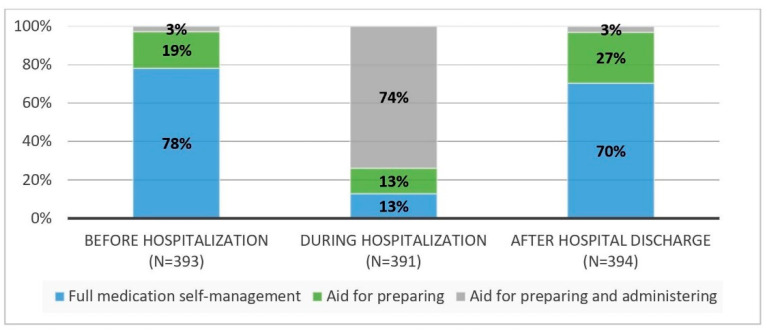
Prevalence of medication self-management before, during and after hospitalization.

**Figure 2 ijerph-18-07031-f002:**
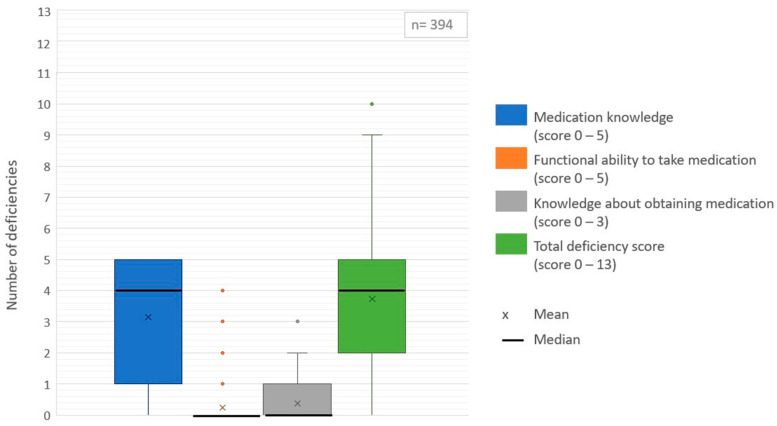
Boxplots of medication management deficiency scores.

**Table 1 ijerph-18-07031-t001:** Sample characteristics (n = 400).

Patient Characteristics		%	Mean [SD]
Age (years) Gender			81.7 [4.97]
Women		52.5	
Men		47.5	
Level of education (EQF ^a^)		(n = 399) ^b^	
None		4.3	
Level 1		21.3	
Level 2/3/4		49.4	
Level 5		11.5	
Level 6		8.0	
Level 7		5.5	
Length of hospital stay			
1 to 3 days		24.5	
4 to 7 days		26.5	
8 to 11 days		21.3	
12 to 15 days		11.5	
≥16 days		16.2	
Reason for hospitalization (multiple answers possible) ^c^			
Treatment		63.9	
Examination		26.7	
Observation		13.5	
Medication review		8.4	
Geriatric risk profile (score 0–6)	(n = 389) ^b^		2.7 [0.95]
Positive geriatric risk profile (≥2/6)		90.7	
Number of chronic diseases			2.1 [1.61]
Number of prescribed medicines at discharge			9.3 [3.38]
Help needed to reside in their own home	(n = 396) ^b^	62.4	

^a^ European Qualifications Framework [30]. ^b^ Deviating sample size due to missing data (missing completely at random) [29]. ^c^ Multiple answers were possible which resulted in a total of more than 100%.

**Table 2 ijerph-18-07031-t002:** Differences between patients who did fully self-manage medication after discharge and patients receiving help with medication management after discharge.

	**Total** n = 393	**SMM ^a^** n = 276	**No SMM ^b^** n = 117	***p* ***
**In-hospital medication management, %**				
I did fully self-manage my medicines during hospitalization.	12.8	14.9	7.8	0.053
I am … in hospital to self-manage my medicines at home.				**<0.001**
Not prepared	43.6	41.2	49.2	
Insufficiently prepared	10.3	6.2	19.8	
Sufficiently prepared	46.1	52.6	31.0	
I stored my medication in my room during hospitalization.^c^	n = 322	n = 227	n = 95	
	19.6	22.9	11.6	**0.019**
I made my own decisions about medication intake during hospitalization. ^c^	n = 329	n = 231	n = 98	
	53.8	57.6	44.9	**0.035**
**Medication-related information at discharge, %**				
I had a conversation on how to manage medicines at home.				0.051
Yes	18.7	18.6	18.8	
No	74.2	76.3	69.2	
I don’t know	7.1	5.1	12.0	
I received … information about medication at discharge.				**0.006**
No	5.6	5.9	5.1	
Too little	31.6	26.5	43.6	
Enough	26.3	26.9	24.8	
Too much	36.5	40.7	26.5	
I received information about … at discharge. ^c,d^	n = 267	n = 193	n = 74	
Working area	33.2	34.0	31.0	0.642
Indication	54.0	55.0	51.4	0.603
Dose	69.3	74.1	56.8	**0.006**
Time	77.9	81.9	67.6	**0.012**
Side effects	9.6	10.5	7.1	0.412
Information about medication was provided … ^c,d^	n = 267	n = 193	n = 74	
Oral	85.8	87.6	81.3	0.189
On paper	55.8	60.1	44.6	**0.022**
The … gave me information about my medicines at discharge ^c,d^	n = 267	n = 193	n = 74	
Physician	40.7	40.8	40.3	0.934
Physician assistant	9.2	8.5	11.3	0.487
Nurse	69.7	74.1	58.1	**0.011**
Other	2.3	2.7	1.4	0.553
**Medication schedule, %**				
I received a medication schedule at discharge.				**<0.001**
No	14.5	15.9	11.2	
Yes and I understand	69.4	78.6	47.4	
Yes and I don’t understand	16.1	5.5	41.4	
The … was mentioned on the medication schedule. ^c,d^	n = 331	n = 232	n = 99	
Medication name	97.9	97.8	98.0	0.938
Working area	8.3	7.5	10.2	0.422
Indication	20.4	19.8	21.9	0.676
Dose	93.7	96.6	86.9	**0.001**
Time	97.9	98.7	95.9	0.108
Side effects	0.6	0.0	2.1	**0.030**
**Medication management after discharge, %**				
Taking responsibility for my medication management after discharge was…………				**<0.001**
Difficult	13.0	5.8	29.9	
Easy	87.0	94.2	70.1	
Medication management assistance …				**<0.001**
Should be extended	13.0	7.6	25.9	
Should remain the same	83.6	88.4	72.4	
Should be restricted	3.4	4.0	1.7	
I stopped the prescribed therapy earlier than agreed with the physician.	22.7	21.1	26.5	0.242
I changed the way medicines should be taken based on own knowledge/experience.	25.4	24.3	28.2	0.413
**Patient satisfaction (mean [SD])**				
Satisfaction with help during hospitalization to self-manage medication after discharge. (scale 0–10)	7.3 [2.32]	7.4 [2.36]	7.1 [2.23]	0.211
Satisfaction with own medication management after discharge. (scale 0–10)	8.2 [1.88]	8.7 [1.16]	6.9 [2.55]	**<0.001**

^a^ SMM: Full self-management of medication after discharge (preparation and administration). ^b^ No SMM: Help with medication management (preparation and/or administration) after discharge. ^c^ This question could only be completed under certain conditions which resulted in a smaller sample size. Some patients had to skip this question based on their answer to a previous closed-ended question. ^d^ Multiple answers were possible which resulted in a total of more than 100%. * *p*-values were calculated using chi-squared tests for nominal and independent t-tests for continuous variables, *p*-values in bold: it indicates the significant values.

**Table 3 ijerph-18-07031-t003:** Logistic regression of predictors of medication management deficiencies (general).

	**Univariate Analysis ^a,b^**				**Multiple Logistic Regression ^e^** (n = 391)
	**Yes ^c^**	**No ^d^**	***p***	**OR [95% CI]**	**OR [95% CI]**
Number of prescribed medicines at discharge (mean [SD])	n = 354	n = 40			
	9.4 [3.42]	7.7 [2.52]	0.002	1.22 [1.07–1.38]	1.19 [1.05–1.35]
In-hospital medication management	n = 352	n = 39			
By nurses (%)	76.1	53.8	0.004	2.74 [1.39–5.37]	2.42 [1.21–4.82]
By patients (full/partial) (%)	23.9	46.2		Ref.	
Patient was … in hospital to manage medication at home	n = 351	n = 39			

Not/insufficiently prepared (%)	55.6	38.6	0.045	2.00 [1.02–3.94]	
Sufficiently prepared (%)	44.4	61.5		Ref.	/

Note: CI = confidence interval; ref = reference, OR = odds ratio; / = these variables were not included in the model. ^a^ All patient and medication management characteristics were studied in univariate analysis. Only significant variables were presented in this table. ^b^ Deviating sample size due to missing data (missing completely at random) [29]. ^c^ Patients with medication management deficiencies after discharge. ^d^ Patients without medication management deficiencies after discharge. e Multiple logistic regression analysis: Nagelkerke R^2^: 0.086, *p* < 0.001. Colored background: This ensures uniformity as deviating n-values have a colored background.

**Table 4 ijerph-18-07031-t004:** Logistic regression of predictors of medication knowledge deficiencies.

	**Univariate Analysis ^a,b^**				**Multiple Logistic Regression ^e^** (n = 390)
	**Yes ^c^**	**No ^d^**	***p***	**OR [95% CI]**	**OR [95% CI]**
Number of prescribed medicines at discharge (mean [SD])	n = 337	n = 57			1.17 [1.05–1.30]
	9.5 [3.45]	7.9 [2.58]	0.001	1.18 [1.07–1.31]	
Geriatric risk profile (score 0–6) (mean [SD])	n = 330	n = 53			/
	2.7 [0.96]	2.4 [0.87]	0.039	1.41 [1.02–1.95]	
In-hospital medication management	n = 335	n = 56			
By nurses (%)	76.4	58.9	0.007	2.26 [1.25–4.07]	/
By patients (full/partial) (%)	23.6	41.1		Ref.	
Patient was … in hospital to manage medication at home	n = 334	n = 56			

Not/insufficiently prepared (%)	56.6	37.5	0.009	2.17 [1.21–3.89]	2.02 [1.12–3.64]
Sufficiently prepared (%)	43.4	62.5		Ref.	

Note: CI = confidence interval; ref = reference, OR = odds ratio; / = these variables were not included in the model. ^a^ All patient and medication management characteristics were studied in univariate analysis. Only significant variables were presented in this table. ^b^ Deviating sample size due to missing data (missing completely at random) [29]. ^c^ Patients with medication knowledge deficiencies after discharge. ^d^ Patients without medication knowledge deficiencies after discharge. e Multiple logistic regression analysis: Nagelkerke R^2^: 0.076, *p* < 0.001. Colored background: This ensures uniformity as deviating n-values have a colored background.

**Table 5 ijerph-18-07031-t005:** Logistic regression of predictors of deficiencies in medication-taking ability.

	**Univariate Analysis ^a,b^**				**Multiple Logistic Regression ^e^** (n = 310)
	**Yes ^c^**	**No ^d^**	***p***	**OR [95% CI]**	**OR [95% CI]**
Age (mean [SD])	n = 58	n = 336			
	83.2 [5.22]	81.5 [4.91]	0.019	1.07 [1.01–1.13]	1.08 [1.02–1.15]
Number of chronic diseases (mean [SD])	n = 57	n = 315			
	2.7 [1.78]	1.9 [1.55]	0.002	1.30 [1.10–1.53]	1.28 [1.07–1.54]
Number of prescribed medicines at discharge (mean [SD])	n = 58	n = 336			
	10.5 [3.93]	9.1 [3.24]	0.004	1.12 [1.04–1.21]	/
Gender	n = 58	n = 336			
Men (%)	34.5	48.4	0.045	0.55 [0.31–0.99]	/
Women (%)	65.5	51.2		Ref.	
Help needed to reside in own home (%)	n = 58	n = 332			
	82.8	58.7	0.001	3.37 [1.65–6.89]	3.83 [1.54–9.52]
Type of hospitalization	n = 58	n = 336			
Unexpected hospitalization (%)	74.1	59.2	0.033	1.97 [1.06–3.69]	/
Planned hospitalization (%)	25.9	40.8		Ref.	
During hospitalization, patient made own decisions about taking medicines, (%)	n = 49	n = 280			
	32.7	57.5	0.002	0.36 [0.19–0.68]	0.38 [0.19–0.75]
Patient was …. in hospital to manage medication at home	n = 57	n = 333			

Not/insufficiently prepared (%)	70.2	51.1	0.009	2.26 [1.23–4.14]	/
Sufficiently prepared (%)	29.8	48.9		Ref.	

Note: CI = confidence interval; ref = reference, OR = odds ratio; / = these variables were not included in the model. ^a^ All patient and medication management characteristics were studied in univariate analysis. Only significant variables were presented in this table. ^b^ Deviating sample size due to missing data (missing completely at random) [29]. ^c^ Patients with deficiencies in medication-taking ability after discharge. ^d^ Patients without deficiencies in medication-taking ability after discharge. ^e^ Multiple logistic regression analysis: Nagelkerke R^2^: 0.187, *p* < 0.001. colored background: This ensures uniformity as deviating n-values have a colored background.

## Data Availability

Data are available from the corresponding author upon reasonable request.

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
