# Peer review of "What Happens after Hospital Discharge? Deficiencies in Medication Management Encountered by Geriatric Patients with Polypharmacy"

_ijerph, 2021, doi:10.3390/ijerph18137031_

Round 1

Reviewer 1 Report

General comment

I commend you for your insight to identify this problem, and for doing research to help practitioners better understand the issues. The topic is very relevant and practice-oriented. The study makes a valuable contribution to the knowledge.

Abstract

line 19-20: report the range of (min, max) post-discharge medication management deficiencies. Also report SD of mean.

Introduction

The introduction is quite brief. Expand the introduction to adequately illustrate why self-management of medication is a problem, and why it is worth studying.  There are mentions of adherence, polypharmacy... define these terms (include citations). A lot of research has been done on medication adherence and medication management. While the unique contribution of your research is its focus on the management of medications by geriatrics in the home, there is more opportunity to discuss the key concepts in your introduction.

line 52-53: Rephrase this. I suggest "...more vulnerable to adverse events from aberrant medication intake." 

line 57-60: consider combining aim 1 and aim 2. I suggest;

1) to evaluate post-discharge self-management of medications by geriatric patients, and 2) to identify factors associated with post-discharge self-management of medications by geriatric patients

(Feel free to rephrase as you like)

line 70-72: you said "convenience sample of wards." Did you sample wards or patients? Clarify.

line 80-81: which method was used to calculate sample size? I see you included a citation, but describe the method in one or two sentences.

line 99-103: Need to rephrase this paragraph. I guess this was an attempt to minimize social desirability bias and nonresponse bias. That's ok. But you need to rephrase the paragraph to make it sound less suspicious. As it is now, it sounds like you did something a little underhanded. Again, I get the idea, but I strongly suggest you rephrase.

line 113-114: I suggest you rephrase. You can say "the lowest possible score was 1 as each participant received 1 point for polypharmacy."

line 119: I suggest "MedMaIDE items were dichotomous as responses were scored Yes (0) and No (1), respectively."

line 120: You said "Only critical terms,..." What does this mean? Explain. Also include in-text citation.

line 124-125: You said "The instrument was administered for the entire medication regimen of the patient." This statement is not clear. Please explain.

line 126-129: Data were collected in addition to MedMaIDE. How was the data collected? Is this different from the "medication knowledge" scale of the MedMaIDE? Please clarify.

line 135: are n-point Likert scales different from 10-point Likert scales? If they refer to the same instrument, you can simply say "10-point Likert scale"

line 138: I suggest you replace "discontinuous" with "categorical". Categorical can be ordinal or nominal.

Results Section

I suggest you reorganize table 1. You can use three columns namely: 1) Participant characteristics, 2) n(%), and 3) mean(SD)

Percentage of men not shown in Table 1

Table 2: superscript d; did you consider how overlap in responses for each category might affect your analysis? The categories of your explanatory variables are not mutually exclusive if the same individual can give multiple responses to the same item. For instance, one patient can give three responses to the item "I received information about ... at discharge." How might this affect your use of chi-square test ? Need to discuss, perhaps in the limitations section.

Table 2: superscript c; what are the conditions being referred to?

line 227: I suggest you report range of medication management deficiencies rather than the maximum. The range is more informative.

line 232: "average of 70%" Is this an average or a percentage? 

line 239: (mean 0.2/5) mean should not be a fraction.

Figure 2: Are these boxplots? Typically, boxplots are used to show medians and interquartile range. You have reported mean (x) in this case. Perhaps it's possible to do that, but I suggest you review this. [You can use bar graph with error bars to show mean and confidence intervals]

Section 3.3: it's not clear how you arrived at your outcome variable. From the text, your outcome variable appears to be medication management deficiencies. But medication management deficiencies have multiple components to it as you have described in previous sections.

When you say in line 262 that "patients with medication management deficiencies took a ...," it appears you have an outcome that is binary (patients with... versus patients without...) which explains the use of logistic regression. That's fine. But you should explain how you arrived at this binary outcome variable. Did you combine all the components of medication management deficiencies? How? What's the rationale?

line 268-270: I suggest you revise the interpretations of OR. Rather than "... increased with 100%", I suggest "patients who were not or insufficiently prepared... were more likely to have (or report) post-discharge medication management deficiencies [OR = 2.00; 95%CI = (1.02, 3.94)]."

Table 3 and 4: What is the purpose of the univariate analyses?

Implications for practice section

Very good points! 

Strengths and Limitations section

line 388-390: I suggest you rephrase or delete "to our knowledge". There have been other studies on this topic and in this area. I suggest you use  terms like "home medication experience" to expand your search.

line 393-394: Other demographic variables like country, race, the use of convenience sampling, etc., will limit generalizability

Other comments

I strongly suggest that you review the statistics. The methods appear to be a good fit, but the procedures can be improved and better described. Also, tables should be improved to make them more reader-friendly.

Overall comment

Research is practice relevant. Well done!

Author Response

Point 1: I commend you for your insight to identify this problem, and for doing research to help practitioners better understand the issues. The topic is very relevant and practice-oriented. The study makes a valuable contribution to the knowledge.

Response 1: Thank you for the motivational feedback.

Point 2: line 19-20: report the range of (min, max) post-discharge medication management deficiencies. Also report SD of mean.

Response 2: Range and standard deviation were added. The maximum of 13 in the original sentence refers to the total number of deficiencies that patients could have according to the MedMaIDE (5 knowledge-related deficiencies, 5 deficiencies related to medication-taking ability and 3 deficiencies related to obtaining medication). This might be confusing with the maximum number of deficiencies that were reported. Therefore, the sentence has been rephrased and the range/SD were added:

“Patients had a mean of four different deficiencies in post-discharge medication management (SD 2.17, range 0-10).”

Point 3:  The introduction is quite brief. Expand the introduction to adequately illustrate why self-management of medication is a problem, and why it is worth studying.  There are mentions of adherence, polypharmacy... define these terms (include citations). A lot of research has been done on medication adherence and medication management. While the unique contribution of your research is its focus on the management of medications by geriatrics in the home, there is more opportunity to discuss the key concepts in your introduction.

Response 3: We expanded the introduction and provided information on the key concepts of the medication self-management process, i.e. the actions required to properly manage medicines. The following text has been added on page 1-2:

The term medication self-management refers to the patient’s need to fulfill a set of actions in order to manage medicines [3]. The process starts with fulfilling a prescription. During the second step, patients should learn how to use medicines in a safe and correct manner which requires medication knowledge. Organizing medication intake and planning daily medication schedules is the third step. The fourth step comprises of actually taking the prescribed medication. Monitoring medication intake and evaluating possible effects or symptoms related to medicines in order to undertake any action if needed is described as the fifth step. The last step concerns the act of sustaining a correct medication intake routine in a safe and appropriate way [3, 4]. However, literature suggests individuals struggle in various ways during the completion of this set of actions influencing medication adherence [5-10]. Adherence can be defined as “the process by which patients take their medications as prescribed” [11].

Definitions of polypharmacy and adherence and citations were added:

  • Polypharmacy is commonly defined as “the minimum concomitant intake of at least five medicines” [16, 17].

Masnoon, N.; Shakib, S.; Kalisch-Ellett, L.;  Caughey, G. E. What is polypharmacy? A systematic review of definitions. BMC Geriatr 2017, 17(1), 230. doi:10.1186/s12877-017-0621-2

Hellemans, L.; Nuyts, S.; Hias, J.; van den Akker, M.; Van Pottelbergh, G.; Rygaert, X.; et al. Polypharmacy and excessive polypharmacy in community-dwelling middle aged and aged adults between 2011 and 2015. Int J Clin Pract 2021, 75(4), e13942.

  • Adherence can be defined as “the process by which patients take their medications as prescribed” [11].

Vrijens, B.; De Geest, S.; Hughes, D. A.; Przemyslaw, K.; Demonceau, J.; Ruppar, T.; . .

     Urquhart, J. A new taxonomy for describing and defining adherence to medications.

     Br J Clin Pharmacol 2012, 73(5), 691-705. doi:10.1111/j.1365-2125.2012.04167.x

Point 4: line 52-53: Rephrase this. I suggest "...more vulnerable to adverse events from aberrant medication intake." 

Response 4:  The sentence has been rephrased: “Moreover, pharmacokinetic and pharmacodynamic changes make the elderly more vulnerable to adverse events from aberrant medication intake.”

Point 5: line 57-60: consider combining aim 1 and aim 2. I suggest; 1) to evaluate post-discharge self-management of medications by geriatric patients, and 2) to identify factors associated with post-discharge self-management of medications by geriatric patients. (Feel free to rephrase as you like)

Response 5:  Aim 1 and 2 were combined. The sentence has been rephrased: “This study aimed (1) to evaluate post-discharge medication self-management and problems encountered by geriatric patients with polypharmacy; and (2) to identify factors associated with post-discharge medication management deficiencies.”

Point 6: line 70-72: you said "convenience sample of wards." Did you sample wards or patients? Clarify.

Response 6: Firstly, we obtained a convenience sample of wards from each hospital, based on the availability of eligible patients. Afterwards, all eligible patients of the selected wards were included during the data collection period until the desired sample size was reached. To select patients, consecutive sampling was used.

To clarify the sampling method of patients in the revised manuscript, the term consecutive sampling was mentioned explicitly:  “All eligible patients of the selected wards, willing to participate, were included during the data collection period until the desired sample size was reached (consecutive sampling).”

Point 7: line 80-81: which method was used to calculate sample size? I see you included a citation, but describe the method in one or two sentences.

Response 7: The following text was added: “The sample size was calculated prior to the study, using the Sampsize calculator for prevalence studies [24]. Precision (%), prevalence (%), target population size and confidence level had to be entered. To reach the highest estimated sample size, an infinite population size (defined by the number 0) and an estimated prevalence of 50% (default value) needed to be used.”

Point 8: line 99-103: Need to rephrase this paragraph. I guess this was an attempt to minimize social desirability bias and nonresponse bias. That's ok. But you need to rephrase the paragraph to make it sound less suspicious. As it is now, it sounds like you did something a little underhanded. Again, I get the idea, but I strongly suggest you rephrase.

Response 8:  The paragraph has been rephrased: “At the day of discharge patients were informed about the fact the study was focusing on discharge management in the broad sense in order to avoid research bias. During the home visit, it was clarified the study focused on medication management in the context of discharge management.”

Point 9: line 113-114: I suggest you rephrase. You can say "the lowest possible score was 1 as each participant received 1 point for polypharmacy."

Response 9: The sentence has been rephrased as suggested.

Point 10: line 119: I suggest "MedMaIDE items were dichotomous as responses were scored Yes (0) and No (1), respectively."

Response 10: The sentence has been rephrased as suggested.

Point 11: line 120: You said "Only critical terms,..." What does this mean? Explain. Also include in-text citation.

Response 11: The sentences have been rephrased to clarify the term critical items and an in-text citation was added:   “Although the MedMaIDE comprises 19 items, only 13 items were used to calculate the deficiency sub scores (per area) and the overall deficiency score. These 13 items are considered critical by Orwig et al. [27], as they are crucial for self-management of medication. These consisted of five medication knowledge-related items, five items related to medication-taking ability and three items related to obtaining medication.”

Point 12: line 124-125: You said "The instrument was administered for the entire medication regimen of the patient." This statement is not clear. Please explain.

Response 12: This statement refers to the medication knowledge-related items of the MedMaIDE. Knowledge was assessed for all medicines a patient had to take (i.e. entire medication regimen). We agree that this was not clearly described. The sentence has been moved to the end of the paragraph and has been rephrased: “Medication knowledge was objectively assessed by research assistants for all patient's medicines based on the medication schedule.”

Point 13: line 126-129: Data were collected in addition to MedMaIDE. How was the data collected? Is this different from the "medication knowledge" scale of the MedMaIDE? Please clarify.

Response 13: In the MedMaIDE, medication knowledge was approached as a dichotomous variable (knowledge yes/no). To answer this dichotomous question, knowledge of each element (name, indication, administration route, time of intake an dose) had to be assessed for all medicines. However, a yes/no question provides only limited information on the extent to which a patient has knowledge of his medicines. That’s why we noted for how many medicines a patient knew the name,… of his/her medicines, so a percentage could be calculated.

We rephrased the sentence to clarify: “The MeDMaIDE uses dichotomous variables to collect data on medication knowledge, which only provides information on whether or not knowledge is present. To gather information on the extent to which a patient has knowledge of his medication, the number of medicines for which the patient knew the name, indication, dose, route of administration and time of intake were noted.”

Point 14: line 135: are n-point Likert scales different from 10-point Likert scales? If they refer to the same instrument, you can simply say "10-point Likert scale"

Response 14: For some questions related to medication management, patients had to rate themselves on a scale from 0 to 10 points. Therefore, we have chosen to use the term 10-point scales as this differs from Likert scales where patients had to indicate to what extent they agree with a statement.

This sentence has been rephrased: “….were surveyed using four- and five-point Likert scales, 10-point rating scales and self-developed multiple choice questions.”

Point 15: line 138: I suggest you replace "discontinuous" with "categorical". Categorical can be ordinal or nominal.

Response 15: We did replace the term.

Point 16: I suggest you reorganize table 1. You can use three columns namely: 1) Participant characteristics, 2) n(%), and 3) mean(SD)

Response 16:  Table 1 has been reorganized as suggested. Three columns were used.

Point 17: Percentage of men not shown in Table 1

Response 17: The percentage of men has been added in table 1.

Point 18: Table 2: superscript d; did you consider how overlap in responses for each category might affect your analysis? The categories of your explanatory variables are not mutually exclusive if the same individual can give multiple responses to the same item. For instance, one patient can give three responses to the item "I received information about ... at discharge." How might this affect your use of chi-square test ? Need to discuss, perhaps in the limitations section.

Response 18: Following text has been added in the limitations section:

“Some questions allowed patients to fill in multiple answers, which resulted in multiple dichotomous variables. Instead of defining multiple response sets, each possible answer was treated as a separate dichotomous variable in the analysis and chi-squared tests were used. However, since the answers of the row variables were not completely independent of each other, the chi-square statistic might have inflated.”

Point 19: Table 2: superscript c; what are the conditions being referred to?

Response 19:  The conditions refer to Question Skip Logic. The following sentence was added in superscript to clarify:

“Some patients had to skip this question based on their answer to a previous closed-ended question.”

Point 20: line 227: I suggest you report range of medication management deficiencies rather than the maximum. The range is more informative.

Response 20: In this sentence, the maximum of 13 refers to the total number of deficiencies that patients could have according to the MedMaIDE (5 knowledge-related deficiencies, 5 deficiencies related to medication-taking ability and 3 deficiencies related to obtaining medication). This might be confusing with the maximum number of deficiencies that were reported. Therefore, the sentence has been rephrased and the range was added:

“Patients had a mean of four different deficiencies in medication management (range 0-10).”

Point 21: line 232: "average of 70%" Is this an average or a percentage? 

Response 21: Both, it is an average percentage. For each patient, we calculated the percentage of medicines for which he/she could correctly identify the time of intake. Afterwards, the mean percentage of all patients was calculated.

Point 22: line 239: (mean 0.2/5) mean should not be a fraction.

Response 22: We no longer presented the mean value as a fraction.

Point 23: Figure 2: Are these boxplots? Typically, boxplots are used to show medians and interquartile range. You have reported mean (x) in this case. Perhaps it's possible to do that, but I suggest you review this. [You can use bar graph with error bars to show mean and confidence intervals]

Response 23: Yes, these are boxplots with indication of the median and mean in order to provide as complete information as possible. Medians are shown as horizontal lines in the figure. However, in boxplot 2 (medication-taking ability) and boxplot 3 (knowledge about obtaining medication) the median is equal to zero and therefore not clearly visible which may be confusing. To clarify this figure, we added the term boxplots in the title. Median was clarified by a symbol in the legend. We improved the visibility of the median lines by coloring them black.

Point 24: Section 3.3: it's not clear how you arrived at your outcome variable. From the text, your outcome variable appears to be medication management deficiencies. But medication management deficiencies have multiple components to it as you have described in previous sections.

Response 24: To clarify the dichotomous outcome variable, the following text has been added to the data analysis section:

“Four binary outcome variables were created: 1) medication management deficiencies in general, 2) medication-related knowledge deficiencies, 3) deficiencies related to medication-taking ability and 4) deficiencies related to obtaining medication. Based on the total deficiency score and sub scores of the MedMaIDE, patients were subdivided into patients with (deficiency score of at least 1) and without deficiencies (deficiency score of 0).”

Point 25: When you say in line 262 that "patients with medication management deficiencies took a ...," it appears you have an outcome that is binary (patients with... versus patients without...) which explains the use of logistic regression. That's fine. But you should explain how you arrived at this binary outcome variable. Did you combine all the components of medication management deficiencies? How? What's the rationale?

Response 25: To clarify the dichotomous outcome variable, the following text was added to the data analysis section:

“Four binary outcome variables were created: 1) medication management deficiencies in general, 2) medication-related knowledge deficiencies, 3) deficiencies related to medication taking ability and 4) deficiencies related to obtaining medication. Based on the total deficiency score and sub scores of the MedMaIDE, patients were subdivided into patients with (deficiency score of at least 1) and without deficiencies (deficiency score of 0).”

Point 26: line 268-270: I suggest you revise the interpretations of OR. Rather than "... increased with 100%", I suggest "patients who were not or insufficiently prepared... were more likely to have (or report) post-discharge medication management deficiencies [OR = 2.00; 95%CI = (1.02, 3.94)]."

Response 26: The sentence has been rephrased:

“Furthermore, patients who were not or insufficiently prepared in hospital to manage medication at home were more likely to have post-discharge medication management deficiencies (OR = 2.00; 95%CI [1.02, 3.94]).”

Point 27: Table 3 and 4: What is the purpose of the univariate analyses?

Response 27: The univariate analysis aimed to identify which variables were significant predictors of medication management deficiencies, because only significant variables from univariate analysis were included in the multiple logistic regression. The following text was added in the data analysis section to clarify the purpose of the univariate analysis:

“First, a univariate analysis was performed to determine which factors were significant predictors of medication management deficiencies.”

Point 28: Implications for practice section Very good points! 

Response 28: Thank you for this positive feedback.

Point 29: line 388-390: I suggest you rephrase or delete "to our knowledge". There have been other studies on this topic and in this area. I suggest you use  terms like "home medication experience" to expand your search.

Response 29: We rephrased the sentence as follows:

“This is the first study examining home medication experience issues and factors related to medication management deficiencies after hospital discharge in a geriatric population with polypharmacy.”

Point 30: line 393-394: Other demographic variables like country, race, the use of convenience sampling, etc., will limit generalizability

Response 30: We agree with this comment. Therefore, we rephrased the sentence:

“Although there was an almost equal proportion of male and female participants (47% vs 53%) which positively affects generalizability of the study results, other demographical variables (e.g. country) and the use of convenience sampling will limit generalizability.”

Point 31:  Other comments: I strongly suggest that you review the statistics. The methods appear to be a good fit, but the procedures can be improved and better described. Also, tables should be improved to make them more reader-friendly.

Response 31: The data analysis was extended to clarify the procedures. The following adjustments have been made.

  • Discontinuous has been replaced by the term categorical.
  • Information on how to treat missing data has been added: “The Little’s MCAR test was used to assess missing values which were found to be missing completely at random (MCAR) [29]. Listwise deletion was used to treat missing cases in each sub analysis.”
  • The following sentences were added to clarify the dichotomous outcome variable used in logistic regression: “Four binary outcome variables were created: 1) medication management deficiencies in general, 2) medication-related knowledge deficiencies, 3) deficiencies related to medication-taking ability and 4) deficiencies related to obtaining medication. Based on the total deficiency score and sub scores of the MedMaIDE, patients were subdivided into patients with (deficiency score of at least 1) and without deficiencies (deficiency score of 0).”
  • We clarified the purpose of the univariate analysis: “First, a univariate analysis was performed to determine which factors were significant predictors of medication management deficiencies.”
  • In addition, we added how multicollinearity was assessed: “Multicollinearity was assessed using correlation matrices.”

We made the tables more reader-friendly through following adjustments:

  • Table 1 is reorganized into three columns: 1) Participant characteristics, 2) %, 3) mean(SD)
  • We have deleted the absolute numbers to make tables more concise. We now only present the percentages.
  • We deleted the columns with test statistics in all tables, as the most important results are described in the text, indicating the test statistics.

Point 32: Overall comment: Research is practice relevant. Well done!

Response 32: Thank you.

Reviewer 2 Report

This study has been adequately conducted. Objectives are clear and relevant. Methods, results and conclusions are linked. I think this is a good study providing interesting and useful information.

However, authors must explain if they have considered the differences using some medicines. The description of the methods does not include this point. Some medicines are more complex than other to be used by the patients. For example, cautions avoiding foods, considering the time of the day, instructions for conservation, ... this aspect is also a relevant factor when this type of studies is been conducted. In my opinion, there is not the same situation at home if more than one patients must organize his/her medicines. In many cases, there is a principal home medication manager who may or may not be the patient being interviewed. If these aspects were not considered, I believe they should be seen as limitations of the study.

Author Response

Point 1: This study has been adequately conducted. Objectives are clear and relevant. Methods, results and conclusions are linked. I think this is a good study providing interesting and useful information.

Response 1: Thank you for this motivational feedback.

Point 2: However, authors must explain if they have considered the differences using some medicines. The description of the methods does not include this point. Some medicines are more complex than other to be used by the patients. For example, cautions avoiding foods, considering the time of the day, instructions for conservation, ... this aspect is also a relevant factor when this type of studies is been conducted. In my opinion, there is not the same situation at home if more than one patients must organize his/her medicines. In many cases, there is a principal home medication manager who may or may not be the patient being interviewed. If these aspects were not considered, I believe they should be seen as limitations of the study.

Response 2: Thank you for this comment. We agree with the fact that some medicines are more complex to manage than others.  However, we assumed that people with polypharmacy have to manage a complex medication regimen anyway because of the higher number of medicines, the higher dosing frequency, interactions, side-effects, et cetera.  Therefore, the main aim was to investigate whether people with complex pharmacotherapy (i.e. polypharmacy) experience medication management deficiencies and which patient and medication management related factors can affect these deficiencies. It would certainly have been interesting to include differences in drug classes in the analyses, but this was beyond the scope of our study. We do, however, recommend further research into differences in medication management between individuals taking different medication(classes).

Therefore, the following text has been added to the limitations section:

“We focused on medication management in patients with polypharmacy (i.e., complex pharmacotherapy) without considering differences in medication management between different medication(classes). This can be seen as a limitation of the study as some medication(classes) are more difficult to manage than others with a possible impact on experiencing medication management deficiencies. Therefore, we recommend to include this aspect in subsequent research.”

Reviewer 3 Report

This manuscript is considered a timely study to improve the safety and compliance of elderly patients with polypharmacy. However, you need to review the data analysis method and some contents of study results as below, so please check.

1) (line 147-149/p.4) Do you know the difference between stepwise logistic regression analysis and multiple regression or multiple analysis? (p.11-12) Also, is the main contents (OR) of Table 3-5 the results of stepwise logistic regression analysis? For continuous variables (age, number of prescribed medicines, geriatric risk profile, etc.), how can the OR be calculated in univariate analysis?

2) Sample's number of all Tables in Results: The total number and the sum of each cell (Table 1) / the number in univariate analysis and the number in multiple analysis (Table 3) are inconsistent, why is that? (For example, N=400, 399 of 'level of education' cell, 446 of 'reason for hospitalization' cell in Table 1). Also, how to treat missing cases in each question (subgroups)? (in Table 2)

3) Please revise the review comments in your original manuscript such as how to write a concise table and consistently describe the references.

Author Response

Answers to reviewer 3

Point 1: This manuscript is considered a timely study to improve the safety and compliance of elderly patients with polypharmacy.

Response 1: Thank you for this positive feedback.

Point 2: However, you need to review the data analysis method and some contents of study results as below, so please check. (line 147-149/p.4) Do you know the difference between stepwise logistic regression analysis and multiple regression or multiple analysis? (p.11-12)

Response 2: We performed a (simple and multiple) logistic regression analysis as our outcome variable is dichotomous (deficiencies yes/no). Simple logistic regression refers to inclusion of only one predictor variable in the analysis while multiple logistic regression includes more than one predictor/independent variable in the analysis. Stepwise regression analysis is one method to perform multiple logistic regression and to select some predictors among all.

We used the terms multiple regression and multiple analysis throughout the paper to distinguish the univariate/simple logistic regression from the multiple logistic regression analysis, meaning the inclusion of several predictors in the logistic regression.  However, the term multiple regression is often used for linear regression. Therefore, we understand the use of this term can be confusing. The term multiple analysis can be confusing as well, as this can be interpreted as performing several analyses one after the other. We have adapted the terms throughout the paper and added the term logistic.

Point 3: Also, is the main contents (OR) of Table 3-5 the results of stepwise logistic regression analysis?

Response 3: First, a simple (univariate) logistic regression analysis was performed to determine which independent variables were significant predictors of medication management deficiencies (yes/no). This is presented in the first columns. Only significant predictors were used in the stepwise logistic regression analysis.  Results of this stepwise logistic regression are shown in the right column. Using the term, ‘multiple analysis’ could be confusing as explained in the previous response. Therefore, we replaced the term ‘multiple analysis/regression’ into ‘multiple logistic regression’ throughout the whole paper.

Point 4: For continuous variables (age, number of prescribed medicines, geriatric risk profile, etc.), how can the OR be calculated in univariate analysis?

Response 4: Using univariate analysis, we assessed the association of the dichotomous outcome variable – medication management deficiencies - with one predictor factor which give us unadjusted ORs. For continuous predictors (e.g., age), the OR represents the increase in odds of the outcome of interest with every one unit increase in the independent variable. For example, in this article: increase in age by one year increases the odds of post-discharge deficiencies in medication-taking ability with 7% (OR of 1.07).

Point 5: Sample's number of all Tables in Results: The total number and the sum of each cell (Table 1) (For example, N=400, 399 of 'level of education' cell, 446 of 'reason for hospitalization' cell in Table 1).

Response 5:  Because of missing data, the sample number of some patient characteristics can differ from the total sample size. For example, only 399 patients reported their level of education (one missing value). To clarify, we explicitly reported deviating sample sizes in table 1 and we added a footnote:

“Deviating sample size due to missing data (missing completely at random).” 

We assessed missing data with the Little’s MCAR test and they were found to be missing completely at random (MCAR). The following sentence was added to the data analysis section to clarify how we treated missing data:

“The Little’s MCAR test was used to assess missing values which were found to be missing completely at random (MCAR) [29]. Listwise deletion was used to treat missing cases in each sub analysis.”

Bennett, D. A. How can I deal with missing data in my study? Aust N Z J Public Health 2001, 25(5), 464-469

In case of ‘reason for hospitalization’: The sum of the cell numbers is higher than the total sample size and, therefore, the percentage is more than 100% as multiple answers were possible. Patients could have been hospitalised for several reasons (e.g. examination and treatment). We have clarified this by adding a footnote:

“Multiple answers were possible which resulted in a total of more than 100%.”

In table 2: For some variables, deviating sample sizes are the result of ‘Skip Logic’. The following footnote was already at the bottom of the table: “This question could only be completed under certain conditions which resulted in a smaller sample size.”  We have added the following sentence to clarify the conditions:  “Some patients had to skip this question based on their answer to a previous closed-ended question.”

Point 6: / the number in univariate analysis and the number in multiple analysis (Table 3) are inconsistent, why is that?

Response 6: In multiple logistic regression analysis, listwise deletion is automatically applied by SPSS. Therefore, the sample size for univariate and multiple logistic regression differs. However, thanks to your comment, we noticed that in some tables related to logistic regression, wrong absolute numbers were reported. Therefore, it was probably not clear why the sample sizes could differ between univariate and multiple logistic regression. The analysis was performed again and we added exact sample sizes per univariate logistic regression analysis within the tables.

Point 7: Also, how to treat missing cases in each question (subgroups)? (in Table 2)

Response 7: The Little’s MCAR test showed data were missing completely at random (MCAR). Listwise deletion was used to treat missing cases in each sub analysis. This sentence was added to the data analysis section.

Point 8:  Please revise the review comments in your original manuscript such as how to write a concise table and consistently describe the references.

Response 8: Thanks to your comment, we have noticed that a few in-text citations were missing. These were added according to the imposed reference style.

We made tables more concise and reader-friendly through following adjustments:

  • Table 1 is reorganized into three columns: 1) Participant characteristics, 2) %, 3) mean (SD)
  • We have deleted the absolute numbers to make tables more concise. We now only present the percentages. However, absolute numbers can be calculated since (deviating) sample sizes were reported in each table.
  • We deleted the columns with test statistics in all tables, as the most important results are described in the text, indicating the test statistics.

Reviewer 4 Report

Thank you for the submission.  The manuscript is interesting. What I found a little odd was the complete lack of mention of pharmacists. Most countries have clinical pharmacists working on hospital wards; discharge counselling and education of patients is one of their routine roles. This not new – it has occurred for many decades in numerous developed countries.  Doesn’t this occur in Belgium? (e.g. Somers A et al. Development of clinical pharmacy in Belgian hospitals through pilot projects funded by the government. Acta Clin Belg. 2019 Apr;74:75-81). Also, a number of countries have systems for post-discharge medication reviews performed by pharmacists in the home of the patient. Again, is this not available in Belgium? Some discussion of the role of clinical pharmacists in Belgium would be useful for international readers.

Lines 48-49: polypharmacy should be defined.

There is no mention of assessing collinearity between variables before proceeding to logistic regression. How was the issue of collinearity/multicollinearity in the multivariate statistical analyses dealt with (assessed and appropriately managed)?  It is likely that some of the variables were inter-related (e.g. age and number of chronic diseases).

Author Response

Answers to reviewer 4

Point 1: Thank you for the submission.  The manuscript is interesting.

Response 1: Thank you for this positive feedback.

Point 2:  What I found a little odd was the complete lack of mention of pharmacists. Most countries have clinical pharmacists working on hospital wards; discharge counselling and education of patients is one of their routine roles. This not new – it has occurred for many decades in numerous developed countries.  Doesn’t this occur in Belgium? (e.g. Somers A et al. Development of clinical pharmacy in Belgian hospitals through pilot projects funded by the government. Acta Clin Belg. 2019 Apr;74:75-81).

Response 2: It is true that pharmacists are not explicitly mentioned in our manuscript. However, we do refer to this healthcare provider indirectly, as we refer to healthcare providers in a broad sense throughout the discussion. In the implications for practice section, we argue that the support of medication self-management by healthcare providers needs to be improved and that healthcare providers can guide patients in this process, for instance through education and counselling. Pharmacists, doctors and nurses should all be involved in this process. This may have been not clear. Therefore, we specified the care providers between brackets:

 “Healthcare providers (i.e. nurses, pharmacists and physicians) can support patients during this process, providing education and counselling.”

Furthermore, in the discussion (paragraph about medication knowledge and education), a few sentences have been added, highlighting the role of the pharmacist.

“However, pharmacists can provide discharge counselling and education as well [36, 37]. Since 2007, pilot projects funded by the government have been launched in more than half of the acute Belgian hospitals to implement clinical pharmacy activities, such as medication counselling at discharge [37].”

Spinewine, A.; Dhillon, S.; Mallet, L.; Tulkens, P. M.; Wilmotte, L.; Swine, C. Implementation of ward-based clinical pharmacy services in Belgium--description of the impact on a geriatric unit. Ann Pharmacother 2006, 40(4), 720-728. doi:10.1345/aph.1G515

Somers, A.; Spinewine, A.; Spriet, I.; Steurbaut, S.; Tulkens, P.; Hecq, J. D.; . . . Haelterman, M. Development of clinical pharmacy in Belgian hospitals through pilot projects funded by the government. Acta Clin Belg 2019, 74(2), 75-81. doi:10.1080/17843286.2018.1462877

Point 3: Also, a number of countries have systems for post-discharge medication reviews performed by pharmacists in the home of the patient. Again, is this not available in Belgium? Some discussion of the role of clinical pharmacists in Belgium would be useful for international readers.

Response 3: Thanks for this comment. Performing a medication review is currently not reimbursed in Belgium, and therefore, not a common practice. However, a new service has been recently introduced, known as the home pharmacist. The following text has been added in the discussion (paragraph about medication knowledge/medication schedule):

“Furthermore, since 2017, patients with polypharmacy can choose a home pharmacist, i.e. a community pharmacist as reference pharmacy. The added value of a home pharmacist consists of the individualized support of these patients and provision of an up-to-date medication schedule [38, 39]. In their systematic review Ensing et al. corroborated the important role of pharmacists to secure continuity of care in multifaceted programs across healthcare settings. Still, they also emphasized that close collaboration with physicians and nurses is essential during and after hospitalization [40]. To stimulate collaboration between general practitioner and pharmacist to improve the safe and rational use of medication, Medical Pharmaceutical Consultations (MPC) can be organized in Belgium. Hospital physicians and hospital pharmacists can also be involved. The main aim of such MPC is to discuss problems in practice and to provide possible solutions to improve pharmaceutical care provided to patients [41].” 

Rijksinstituut voor Ziekte- en Invaliditeitsverzekering. Verklarende nota bij de 37ste wijzigingsclausule bij de overeenkomst tussen de apothekers en de verzekeringsinstellingen. Brussel 2017. Available online: https://www.riziv.fgov.be/SiteCollectionDocuments/overeenkomst_apothekers_Wijzigingsclausule37.pdf (accessed 19 June 2021)

Robberechts, A.; De Petter, C.; Van Loon, L.; Rydant, S.; Steurbaut, S.; De Meyer, G.;  De Loof, H. Qualitative study of medication review in Flanders, Belgium among community pharmacists and general practitioners. Int J Clin Pharm 2021. doi:10.1007/s11096-020-01224-9

Ensing, H. T.; Stuijt, C. C.; van den Bemt, B. J.; van Dooren, A. A.; Karapinar-Çarkit, F.; Koster, E. S.; Bouvy, M. L. Identifying the Optimal Role for Pharmacists in Care Transitions: A Systematic Review. J Manag Care Spec Pharm 2015, 21(8), 614-636. doi:10.18553/jmcp.2015.21.8.614

Koninklijk besluit tot vaststelling van de voorwaarden en nadere regels waaronder het medisch-farmaceutisch overleg wordt toegepast en tot wijziging van het koninklijk besluit van 3 juli 1996 tot uitvoering van de wet betreffende de verplichte verzekering voor geneeskundige verzorging en uitkeringen, gecoördineerd op 14 juli 1994. Belgisch Staatsblad 2015. Available online:

       https://www.ejustice.just.fgov.be/cgi_loi/change_lg.pl?language=nl&la=N&cn=2015040312&table_name=wet (accessed 21 June 2021)

Point 4: Lines 48-49: polypharmacy should be defined.

Response 4: Polypharmacy was defined in line 47-48. However, it may not have been clear that this was a definition. Therefore, we stated it more explicitly.

Polypharmacy is commonly defined as “the minimum concomitant intake of at least five medicines” [16, 17].

van den Akker, M.; Vaes, B.; Goderis, G.; Van Pottelbergh, G.; De Burghgraeve, T.; Henrard, S. Trends in multimorbidity and polypharmacy in the Flemish-Belgian population between 2000 and 2015. PloS one 2019, 14(2), e0212046. doi:10.1371/journal.pone.0212046

Masnoon, N.; Shakib, S.; Kalisch-Ellett, L.; Caughey, G. E. What is polypharmacy? A systematic review of definitions. BMC Geriatr 2017, 17(1), 230. doi:10.1186/s12877-017-0621-2

Point 5: There is no mention of assessing collinearity between variables before proceeding to logistic regression. How was the issue of collinearity/multicollinearity in the multivariate statistical analyses dealt with (assessed and appropriately managed)? 

Response 5: Multicollinearity was assessed using correlation matrices. However, we did not found strong correlations between variables. Some examples:

  • No relationship was found between age and number of chronic diseases (r= -0.007; p = 0.886).
  • No relationship between age and number of prescribed medicines (r= -0.022, p = 0.662)
  • Weak correlation between age and geriatric risk profile score (r= 0.127, p = 0.012)

If we had found strong relationships between variables, we would have included only the variable with the highest significance in the multiple logistic regression. But this was not the case.

The following information was added to statistical analyses:

“Multicollinearity was assessed using correlation matrices.”

Point 6: It is likely that some of the variables were inter-related (e.g. age and number of chronic diseases).

Response 6: No relationship was found between age and number of chronic diseases (r= -0.007; p = 0.886). Therefore, both variables were included in multiple logistic regression analysis.

Round 2

Reviewer 2 Report

Authors have substantially improved the manuscript